# Awareness and compliance with tobacco control policies among retailers near schools in Arghakhanchi, Nepal: A mixed- methods study

**Arjun Poudel**[1]*, **Hari Prasad Kaphle**[1], **Sunita Poudel**[1], **Sagar Parajuli**[1], **Kumud Bhurtel**[2], **Shalik Ram Dhital**[3]

**1** School of Health and Allied Sciences, Pokhara University, Pokhara, Kaski, Nepal, **2** Central Department of Public Administration, Tribhuvan University, Kathmandu, Nepal, **3** Home in Place, Newcastle, Australia

* poudel.arjun28@gmail.com

## Abstract

Tobacco consumption is the second leading cause of death among adults, and it is a major public health challenge worldwide including Nepal. This paper aims to assess awareness and compliance of tobacco legislation among retailers located nearby schools, and facilitators, and barriers to the implementation of legislation at the district level. A cross-sectional mixed-methods study design was carried out among randomly selected retailers (n = 203) located within 100 meters of schools from 51 sites, along with 8 Key Informant Interviews (KII) with purposefully selected implementation-level officials. Data were collected using an interview schedule, a checklist, and KII guideline. Awareness and compliance of retailers were measured for 14 provisions of the Tobacco Products (Control and Regulatory) Act (TPCRA), 2011 of Nepal. Bivariate and multivariate analyses were conducted to identify associations between dependent variables and independent predictors for policy compliance, and manual thematic analysis was performed for qualitative data. Only one third (33%) of retailers were found aware on provisions of TPCRA while majority (93.6%) of retailers were found violating policy provision by selling tobacco close to school. Only 14.7% of retailers were found in compliance with provision of not selling tobacco to minors and 5.8% retailers were found restricting sale of loose cigarettes. Retailers of rural settings were found two times (aOR 2.50, 95% CI, 0.99-6.27) more likely to comply with banning to tobacco sale to minors as compared to urban retailers. The enforcement of legislation, dedicated inspection mechanism and raised public awareness were found as key facilitators for policy implementation. Awareness and compliance among retailers were found to be inadequate despite the enforcement of the TPCRA since 2011. In order to strengthen policy implementation, it is essential to raise awareness, conduct regular and rigorous market inspection, and empower officials and elected representatives to take proactive measures.

**Data availability statement:** All relevant data are within the paper and its Supporting Information files.

**Funding:** The author(s) received no specific funding for this work.

**Competing interests:** The authors have declared that no competing interests exist.

## Introduction

Tobacco is a global public health threat, leading to non-communicable diseases and it creates high social, economic, and environmental burdens [1,2]. In Nepal, adults consuming tobacco was 28.9% in 2019 and their average age of initiation of tobacco consumption was 17.9 years [3,4]. The quite consistent rate (28%) of tobacco consumption prevalence among adults was found in 2022 [5]. Approximately, 27,100 deaths a year were attributed to diseases related to tobacco use (14.9% of all deaths) and, at the same time cigarette smoking and tobacco consumption including vaping is also increasing in school going children [6]. Likewise, 9% of school-going children (aged 13–17 years) were current tobacco users, and among them, the average age of early initiation of tobacco consumption was 14 years [7]. The direct medical costs for treating tobacco-related illnesses account for 5.3% of India's annual private and public health spending, and it also impacts more than 1% of Gross Domestic Product (GDP) of India [7]. Similarly, adult using tobacco is quite similar in Nepal and India; the economic impacts of tobacco use in Nepal may also be comparable to that of India.

Nepal signed the World Health Organization (WHO) Framework Convention on Tobacco Control (FCTC) on December 3, 2003, and ratified it on November 7, 2006 [8,9]. After its ratification, the Government of Nepal enforced the TPCRA in 2011, which includes various legal provisions to restrict promotion and advertisement of tobacco, reduce access to tobacco products, and ban both smoking and use of smokeless tobacco in public places [10]. The Government of Nepal targeted to reduce tobacco consumption from 30.8% to 15% at the end of Sustainable Development Goals (SDG), however despite the implementation of strong tobacco control policies and regulations, the prevalence of tobacco consumption is still high [11–13].

The enforcement of laws is the best strategy to reduce access [14] and aims to safeguard and enhance public health by implementing diverse strategies to decrease tobacco consumption. Since the enactment of tobacco legislation in Nepal, the tobacco industries have consistently interfered, mainly through lobbying with political parties and high level bureaucratic officials, and filed several cases in the Supreme Court against the government to delay or weaken the implementation of tobacco control legislation [15]. The understanding of the compliance of tobacco control laws with the protection of minors needs scientific evidence to provide policy options and realistic scenarios for national and local authorities [16]. This study was therefore carried out among the retailers who are key stakeholders in tobacco products supply system to minors and are also the vital policy implementers at the district level. Laws and regulation are enforced for reduction of tobacco access and use, but laws are ineffective if not properly implemented. This study helps to provide scientific insights and evidence to provide policy guidance and realistic scenarios for authorities at different level by assessing retailers' awareness and compliance with tobacco laws, and investigating the factors that facilitate and hinder their implementation at the district level.

## Materials and methods

### Ethics statement

The ethical approval for the study was obtained from the Institutional Review Committee, Pokhara University, Nepal (Ref. No. 190/2080/081) and prior approval was taken from National Health Education, Information and Communication Centre (National Tobacco Control Focal point) of Nepal. Data collection was done after following verbal and written consent from participants.

### Study design, setting and participants

A cross-sectional study design was carried out among retailers at three local levels namely Sitganga and Sandhikharka Municipality, and Chatradev Rural Municipality of Arghakhanchi district in Nepal. Data collection took place from May 16, 2024, to July 01, 2024. Quantitative data were collected from retailers located within 100 meters of school, while qualitative data were taken from district administration officers, local level tobacco inspectors, and school administrators as they are the key tobacco policy implementers in the district.

### Sample size and sampling procedure

Sample size calculation was done taking the proportion of compliance with tobacco legislation based on previous findings of similar studies [17]. The sample size was 190 calculated by applying Cochran's formula [18]. Three out of 6 local levels were randomly selected from the district, and 51 school sites were proportionally and randomly chosen from 65 eligible secondary level (both public and private) schools. At each site, five retailers were initially selected using a lottery technique in the field. If fewer than five eligible retailers were identified within 100 meters of a school, all available retailers were included in the sample. Participants for KII were purposively selected among the key officials to gather qualitative information, such as assistant chief district officers at district level and chief executive officers of each local level. They are assigned as tobacco inspectors as per law. In addition to them, the district chief of police and school headteacher were taken, as they are crucial persons to enforce the tobacco legislation close to schools.

### Data collection tools and procedure

This study used an interview schedule and observation checklist, adopted from previous similar types of studies [2,16,19–22] and which was modified to the Nepalese context. The tools were pretested among retailers residing close to school and experts were consulted to ensure the reliability and validity of tools. The experts working at the national tobacco control focal point of Nepal were consulted to ensure that the required provisions of TPCRA were incorporated into the observation checklist and interview schedule. The needful revisions were made based on feedback from pretesting and experts' consultation before finalization. A total of 206 eligible retailers across 51 school sites were approached, of which 203 (owners or employees) agreed to participate in the study. The precise distance of each retail store from the schools was measured using a mobile application.. The number of key informant interviews was determined based on the principle of saturation [23]. Interviews were conducted at the official's respective workplaces using an interview guide and were recorded using a recording device.

### Study variables

**Independent variables.** The background characteristics (age, sex, ethnicity, educational level, types of retail stores, settings, duration of business, knowledge regarding harms of tobacco, tobacco consumption behavior, and distance from school) of retailers and the overall awareness on provisions and attitude of respondents towards tobacco control legislation were included as the independent variables for this study.

**Dependent variables.** Compliance (yes/no) of 14 different provisions of TPCRA was measured for awareness and compliance with legislation but among them, only 3 provisions those are directly related to restrict the tobacco consumption among minors were taken as outcome variables in regression analysis.

## Data analysis

The data were entered in Epi-data software and transferred into SPSS version 22.0 for further data analysis. After cleaning and coding the data, the descriptive analyses were initially performed to summarize all background variables and assess retailers' awareness towards TPCRA provisions by estimating percentages. Similarly, compliance with 14 provisions of the TPCRA was analyzed by calculating frequencies and percentages. The Chi square test was then conducted to explore associations between independent variables and all the 14 provisions of TPCRA. Those three key rules of the TPCRA associated with the independent variables with a p-value <0.25 in bivariate analysis were included in regression analysis, where adjusted odds ratios (AOR) with 95% confidence intervals were computed using SPSS version 22.0.

Concurrently, qualitative data were manually analyzed through thematic analysis, involving coding transcripts, identifying subthemes, and generating overarching themes. Initially, audio recordings of the interviews were transcribed, and then transcripts were translated into English. After translation of the transcripts, all authors engaged to read and re-read several times to generate the basic idea about facilitators and barriers to the implementation of tobacco control policies. Then, initial codes were identified in each line of the transcripts, and the codes were analyzed and grouped into similar categories, which were later assembled under subthemes and theme.

## Results

As shown in Table 1, this study included 49 rural and 154 urban retailers, and among them, 128 were groceries and 75 were eateries. Less than half (43.8%) retailers were located within 50 meters of the school against the rule, among them 88.8% retailers were selling tobacco products at restricted places. Almost all of the retailers were convinced that tobacco harms health, still, one-third of them consumed tobacco within last 30 days.

Further, the awareness of legislation was highest regarding ban of smoke in public places (77.8%) and restriction of advertisement of tobacco product (74.6%). Despite having such awareness, most of the participants are still selling the tobacco products nearby school territory. Less than half of the retailers (40.9%) were found aware of the provision restricting tobacco sales to minors, and around one-third (37.3%) were convinced on prohibition of tobacco products sales within 100 meters of a school, but still, most of the participants (90.2%) are still selling the tobacco product to minors at school areas. At the same time, only a few (9.7%) retailers were found informed on prohibition of the sale of loose or stick cigarette by the laws, while the majority of them (73%) were aware of the provisions of penalties in cases of violation of tobacco control laws. Overall, one-third (33.0%) of the retailers were found aware of rules of tobacco control legislation but most of the participants (91.8%) are still selling the tobacco within 100 meters of school (table 2).

Table 3 shows that the observed compliance of displaying the notice "smoking is prohibited" was 5.8% and the displaying signage on "minors are not allowed to buy and sell" at the retail store was null. Additionally, the study indicated that 79.5% displayed tobacco products and the majority (85.3%) sold tobacco to minors where both groceries and eateries almost equally violated the rule. Similarly, 94.2% of retailers sold loose cigarettes, and 93.6% of retailers sold tobacco within 100 meters of school in a prohibited area by violating the laws.

Table 4 shows that, the odds ratio of respondents with an unfavorable attitude towards the legal provisions of the TPCRA was 6.67 times more likely to comply with one of the provisions that prohibits the decoration of retail store with tobacco products. Moreover, the odds of complying with the provision that prohibits the sale of tobacco to minors aged below 18 were twofold (aOR 2.50, 95% CI 0.99–6.27) greater among rural retailers compared to urban retailers. Likewise, the odds ratio of respondents with an unfavorable attitude towards the legal provisions of the TPCRA was five times more

**Table 1. Background characteristics of retailers.**

| Characteristics | Responses | Frequency (N = 203) | Retailers nearby school | |
|---|---|---|---|---|
| | | | Not selling tobacco (n = 13) | Selling tobacco (n = 190) |
| | | n (%) | n (%) | n (%) |
| Sex | Male | 87 (42.9) | 4 (4.6) | 83 (95.4) |
| | Female | 116 (57.1) | 9 (7.8) | 107 (92.2) |
| Ethnicity | Disadvantaged | 43 (21.2) | 9 (7.0) | 119 (93.0) |
| | Advantaged | 160 (78.8) | 4 (5.3) | 71 (94.7) |
| Education level | Up to basic school | 111 (54.7) | 6 (5.4) | 105 (94.6) |
| | Secondary or above | 92 (45.3) | 7 (7.6) | 85 (92.4) |
| Study setting | Rural | 49 (24.1) | 1 (2.0) | 48 (98.0) |
| | Urban | 154 (75.9) | 12 (7.8) | 142 (92.2) |
| Type of retailer | Groceries | 128 (63.1) | 9 (7.0) | 119 (93.0) |
| | Eateries | 75 (36.9) | 4 (5.3) | 71 (94.7) |
| Type of nearby school | Basic school | 55 (27.1) | 5 (9.1) | 50 (90.9) |
| | Secondary school | 148 (72.9) | 8 (5.4) | 140 (94.6) |
| Retailers distance from school | Up to 50 meters | 89 (43.8) | 10 (11.2) | 79 (88.8) |
| | 51-100 meters | 114 (56.2) | 3 (2.6) | 111 (97.4) |
| Aware on harms of tobacco | Yes | 193 (95.1) | 13 (6.7) | 180 (93.3) |
| Use of tobacco by retailer | Yes | 53 (26.1) | 1 (1.9) | 52 (98.1) |
| Duration of retail business (n = 190) | 1-10 years | 123 (64.7) | 0 | 123 (100.0) |
| | 11 or above | 67 (35.3) | 0 | 67 (100.0) |
| Inspection in the last 12 months | Yes | 145 (71.4) | 8 (5.5) | 137 (94.5) |
| Ever been fined for violation of rules (n = 145) | Yes | 24 (16.6) | 1 (4.2) | 23 (95.8) |

**Table 2. Awareness of legislation provisions among retailers.**

| Provisions of TPCRA | | (n = 185) % | Selling of tobacco products in nearby retailers (within 100 meters from school) | |
|---|---|---|---|---|
| | | | No n (%) | Yes n (%) |
| Smoking is banned at public places | | 144 (77.8) | 9 (6.3) | 135 (93.8) |
| Must display of public notice on "smoking is prohibited" at retail store | | 79 (42.7) | 8 (10.1) | 71 (89.9) |
| Advertisement of tobacco is restricted | | 138 (74.6) | 7 (5.1) | 131 (94.9) |
| Restrict to sale tobacco to minors (n = 149) | | 61 (40.9) | 6 (9.8) | 55 (90.2) |
| Restrict to sale within 100 meters of school | | 69 (37.3) | 3 (4.3) | 66 (95.7) |
| Sale of loose cigarettes is prohibited | | 18 (9.7) | 3 (16.7) | 15 (83.3) |
| Aware on provision of penalty in case of violation | | 135 (73) | 12 (8.9) | 123 (91.1) |
| Attitude towards provisions | Unfavorable | 96 (47.3) | 7 (7.3) | 89 (92.7) |
| | Favorable | 107 (52.7) | 6 (5.6) | 101 (94.4) |
| Overall Awareness | Unaware | 124 (67.0) | 5 (4.0) | 119 (96.0) |
| | Aware | 61 (33.0) | 5 (8.2) | 56 (91.8) |

**Table 3. Compliance of retailers with legislative provisions by retail type.**

| Provisions of TPCRA | Overall Compliant (n = 190) % | Compliant | |
| --- | --- | --- | --- |
| | | Groceries (n) % | Eateries (n) % |
| 1) Absence of active smoking at retail store | 129 (67.9) | 86 (66.7) | 43 (33.3) |
| 2) Openly unavailable of lighter or matchstick at store | 103 (54.2) | 72 (69.9) | 31 (30.1) |
| 3) Displaying of public notice on "smoking is prohibited" | 11 (5.8) | 2 (18.2) | 9 (81.8) |
| 4) Displaying of signage on " minors are not allowed to buy and sell " at retail store | 0 (0.0) | – | – |
| 5) No displaying of hoarding boards for advertisement | 180 (94.7) | 112 (62.2) | 68 (37.8) |
| 6) Non decorating of retail store for attraction with tobacco items | 135 (71.1) | 82 (60.7) | 53 (39.3) |
| 7) No displaying of tobacco products at retail store | 39 (20.5) | 24 (61.5) | 15 (38.5) |
| 8) Not allowing to smoke at retail store | 34 (17.9) | 23 (67.6) | 11 (32.4) |
| 9) No advertising or promoting of store from any medium | 176 (92.6) | 110 (62.5) | 66 (37.5) |
| 10) No selling of tobacco products to minors | 28 (14.7) | 16 (57.1) | 12 (42.9) |
| 11) No selling of loose or stick cigarettes | 11 (5.8) | 7 (63.6) | 4 (36.4) |
| 12) Never receiving a coupon from tobacco companies | 184 (96.8) | 113 (61.4) | 71 (38.6) |
| 13) Not distributing of tobacco products at free of cost | 128 (76.4) | 87 (68.0) | 41 (32.0) |
| 14) Not selling tobacco within 100 meters of school (N = 203*) | 13 (6.4) | 9 (69.2) | 4 (30.8) |

* Total sample size

likely to comply with the provision that restricts the sale of loose cigarette compared to respondents with a favorable attitude (Table 4).

Table 5 shows that the enforcement of TPCRA, assigned inspection authorities and raised public awareness on harms of tobacco consumption were found as facilitators for execution the tobacco control legislation. The lack of resources and orientation for market inspection and monitoring activities, unaware retailers on legal provisions, weak and irregular market inspection and poor implementation of TPCRA were found to be barriers for proper implementation of the tobacco control policies in Nepal.

## Discussion

The key findings of this study indicate a significant lack of awareness about tobacco control regulations among retailers, frequent violations of many provisions, and inadequate enforcement of tobacco legislation. Notably, only one-third of the retailers reported being aware of the various provisions of TPCRA of Nepal. Similar studies conducted in Pakistan, the United States, and several cities in India have also reported that poor awareness among retailers regarding tobacco control legislation in their respective countries, underscoring a common challenge in implementing tobacco control policies effectively [19,24–26]. In contrast with these finding, a study done in Mumbai, India, found greater awareness of legal provisions [27]. This indicates that awareness among tobacco retailers of the provisions of tobacco legislation is poor in many countries around the world, be it developing countries from South-Asia or developed countries like USA. Additionally, even in a single country, the level of awareness varies among different cities. This shows that, there is need of conducting more qualitative studies to understand this difference in different areas around the world.

This study revealed that more than one-third of the respondents were convinced that the sale of tobacco products within 100 meters of a school is prohibited. A study done in India also found limited awareness on the same context [19], but in contrast with these findings, other studies conducted in Kathmandu, Nepal, and Mumbai in India found higher consciousness among retailers [16,17,20]. The variation could be a result of the differences in the study setting, such as some studies conducted in rural areas and others in urban areas. Likewise, we found the large majority of retailers were alert to restricting the sale of tobacco to minors aged below 18 years. Different studies conducted in India found similar

**Table 4. Predictors of retailer's compliance with the legislation.**

| Variables | Not decorating stores with tobacco items | Not selling of tobacco to minors | Not selling loose cigarettes |
|---|---|---|---|
| | aOR (95% CI) | aOR (95% CI) | aOR (95% CI) |
| **Distance to school from retail store** | | | |
| Up to 50 meters | 1.51 (0.39-5.82) | 1.10 (0.45-2.69) | 0.66 (0.17-2.53) |
| 51-100 meters (ref) | – | – | – |
| **Tobacco use by retailer** | | | |
| No | 0.87(0.18-4.04) | 1.86 (0.60-5.77) | 1.14 (0.24-5.29) |
| Yes (ref) | – | – | – |
| **Type of nearest school** | | | |
| Secondary school | 1.93 (3.52-10.65) | 0.55 (0.20-1.52) | 1.93 (0.35-10.65) |
| Basic school (ref) | – | – | – |
| **Overall awareness** | | | |
| Aware | 0.51 (0.11-2.26) | 0.88 (0.34-2.30) | 0.51(0.11-2.26) |
| Unaware (ref) | – | – | – |
| **Duration of retail business (years)** | | | |
| 1-10 | 0.95 (0.19-4.64) | 1.96 (0.68-5.62) | 1.04 (0.21-5.11) |
| 11 or above (ref) | – | – | – |
| **Educational level** | | | |
| Secondary or above | 3.41 (0.79-14.64) | 1.48 (0.57-3.79) | 3.41 (0.79-14.64) |
| Up to basic level (ref) | – | – | – |
| **Attitude** | | | |
| Unfavorable | 6.67(1.27-34.95) * | 1.22 (0.50-2.99) | 0.15 (0.29-0.78) * |
| Favorable (ref) | – | – | – |
| **Study setting** | | | |
| Rural | 0.20 (0.24-1.83) | 2.50(0.99-6.27) * | 4.81(0.54-42.48) |
| Urban (ref) | – | – | – |
| **Ethnicity** | | | |
| Advantaged | 1.05 (0.19-5.77) | 3.90 (0.83-18.23) | 1.05(0.19-5.77) |
| Disadvantaged (ref) | – | – | – |
| **Type of retailer** | | | |
| Groceries | 1.07 (0.27-4.23) | 0.78 (0.30-1.98) | 0.93 (0.23-3.69) |
| Eateries (ref) | – | – | – |

*Statistically significant (p<0.05)

findings [16,17,19] but only 21.3% retailers were aware regarding the similar legislation in Pakistan [11]. This might be because the participating retailers were not from nearby educational institutions in Pakistan, and they might be less focused on the provision of sales to minors, which could be the reason for the dissimilarity in findings.

Regarding compliance, large number of retailers violated the rule by selling tobacco close to schools, which is quite consistent with different studies conducted in Saudi Arabia, Nepal and China [13,20,21]. These similarities could be due to the poor enforcement of tobacco legislation in most countries. Furthermore, we found that more retailers sold tobacco products to minors than the eateries, similar to the previous findings from India and Kenya [19,28–31], but contrasting with the results of Mumbai, India, and Thailand [27,31]. The similar results may be attributed to weak enforcement and lower socioeconomic status, while the differences could be due to the fact that India has already implemented tobacco control legislation. This finding suggests that the level of compliance with restrictions on the sale of tobacco to minors is generally

**Table 5. Barriers and facilitators for implementation of tobacco control policies.**

**Theme: Facilitators**

| Subthemes | Description | Sample of Representative Verbatim Quotes |
|---|---|---|
| Enforcement of laws | All of the personnel agreed that enforcement of Act is a significant step to regulate tobacco control in Nepal. | *"The current enforced law is sufficient to be banned and regulated, but it is necessary to know the things that should be revised from time to time."* District administrator1 |
| Assigned inspection authorities to inspect market | Most of the personnel reported having the nominated inspection mechanism at the district and local level, which is the significant step to regulate effectively. | *"The government has also assigned me and the chief administrative officer of the local level as an inspection officer. We have advanced the monitoring in the market frequently."* District administrator 1 |
| Increased public awareness on harms of tobacco consumption | All of the officials shared that raising awareness among the public and retailers is one of the most significant enablers for tobacco control. | *"We have also been organizing various awareness activities in schools, to discourage them to have tobacco. From which I feel that the students are becoming aware."* School administrator1 |

**Theme: Barriers**

| Subthemes | Description | Example of Symbolic Verbatim Quotes |
|---|---|---|
| Unawareness among retailers on provisions | The most of the officials noted that because of insufficient information about the legal provision for retailers, itis very less likely to comply with the rule. | *"An orientation program for the shopkeepers is needed; they are unknown regarding the tobacco legislation due to that it is challenging for implementation."* District Administrator 3 |
| Lack of resources to regulate the laws | The officials mentioned that the scarcity of resources and programs hinders the tobacco control activities at the district level. | *"We no longer have a program and resources to orient them to implement this law."* District administrator 3 |
| Lack of orientation on legal provisions to regulating personnel | Regulating personnel were deprived of the information regarding legal provisions and approaches to tobacco control, and they agreed that better orientation would enhance the implementation of tobacco control policies. | *"I have not participated in such a designated tobacco control legislation orientation program till yet."* Local Tobacco Inspector 1 |
| Retailers' disobedience of enforcement | School administrators have observed tobacco retailers selling tobacco within school premises, who have disregarded their appeal. | *"There are retail grocery stores near our school, sell tobacco products and some retailers did not follow our request."* School administrator 2 |
| Weak implementation and lack of scheduled inspection mechanism | The administrators identified a lack of proper scheduled tobacco specific inspection at tobacco retail stores. The inspection is rare and done only during general market inspections, particularly only during festivals. | *"Market inspection and monitoring also happens sometimes in two years, sometimes in one year, it is not regular."* School Administrator 2 <br> *"It has not been done so far to regulate tobacco products as a separate dedicated type of inspection, only done during regular market monitoring."* Local Tobacco Inspector 2 |

higher in developed countries and cities compared to developing nations. The latest data is unavailable to explore the tobacco prevalence among school-going children in Nepal; the regular youth tobacco prevalence survey is needed.

The results of violating the rule by not displaying a public notice on "Tobacco consumption is prohibited" and "Minors are not allowed to buy and sell tobacco products" at retail stores are highly consistent with the results of previous studies [2,20,26,32–34]. The findings highlight the poor enforcement of tobacco control policies, particularly in developing countries. However, in contrast to findings from other studies [21,25,35] suggest that enforcement and compliance may be relatively better in large cities. This indicates a significant lack of public notices at retail stores in most underdeveloped areas.

Previous studies [21,25,30,36] also revealed poor compliance with the non-display of tobacco products at the point of sale, as per laws, in accordance with this study. The level of compliance was higher in relatively developed nations [24,32,34] which could be due to better enforcement and awareness, suggesting that retail stores are being attracted to minors by the display of tobacco products. Moreover, we reported the highest (94.2%) retailers violating the rule by selling loose or stick cigarettes to customers, which is consistent with the previous studies [30,37,38] which indicates poor

 

enforcement of rules in similar settings and sociocultural contexts. However, few studies from Saudi Arabia and Thailand found contrasting results [32,34]. The variation might be due to the types of retail stores included in the survey. This suggests that minors are easily able to get loose cigarettes, which helps them consume tobacco with less pocket money.

The study found that the respondents with an unfavorable attitude towards the legal provision regarding banning to decorate the retail store with tobacco items was six times more likely to comply of not decorating tobacco products in retail stores, while aware vendors were four times more likely to violate the rule by promoting tobacco items in Pakistan [19]. These divergent findings might be due to differences in the legal provisions of penalties and different cultural backgrounds between Nepal and Pakistan. This indicates that awareness alone, without proper enforcement of legislation, cannot compel retailers to follow the rules.

Moreover, the odds of complying with the rule that no one should sell tobacco to minors are two times greater among rural retailers; similar to these findings, rural vendors are less likely to sell tobacco to minors in Pakistan [19], and there is no association between the type of tobacco retail store and compliance with a minor's tobacco sales ban, similar to the study done in Mumbai, India [27]. The vendors located in rural areas are expected to follow the rules, which may be due to higher community surveillance and familiarity between retailers and minors but the type of retail store doesn't impact them.

Similar to studies conducted in Kenya and India, this research found that key factors facilitating the implementation of tobacco control policies include the enforcement of regulations, market inspection authorities, decentralized power, local commitment, public awareness, and supportive retailers [39–41]. These emerging countries share socioeconomic similarities, and these facilitators play a crucial role in successfully implementing tobacco control legislation. Effective execution of specific acts, raising public awareness, defining control mechanisms at various levels, and demonstrating commitment are all essential for the successful implementation of tobacco control policies.

This study reveals barriers to legislation implementation, including unawareness among retailers, underpowered school authorities, non-capacitated officials, scarcity of resources, poor enforcement, interference from tobacco companies, allowing production for tax collection, and retailer's objections to following rules which are similar to other studies [39–42]. Kenya faces challenges due to the unclear roles of tobacco inspectors, whereas Nepal's legislation includes clear provisions for regulation in this regard [40]. In low- and middle-income countries, poor socioeconomic conditions, including a lack of awareness, create barriers that hinder the effective enforcement of tobacco control legislation. To overcome these challenges, a new health promotion approach, along with a shift in mindset among stakeholders and the government, must be established in alignment with global best practices in tobacco regulation and policy implementation.

This study's reliance on a single shot of observed and self-reported data introduces potential biases. At the same time, retailers might hide their illegal practices intentionally because of fear of legal action, even if they were well informed that the data were only to be utilized for study purposes. This might lead to social desirability bias. These biases can be regarded as limitation of the study which could impact the actual compliance of retailers towards legal provisions.

## Conclusion

This study highlights significant gaps in both awareness and compliance with tobacco control legislation among retailers. It reveals limited knowledge regarding key restrictions, such as the sale of tobacco products within 100 meters of schools and the prohibition of selling tobacco to minors. Additionally, a high density of tobacco retailers at restricted place and widespread non-compliance with many provisions, including the sale of loose cigarettes and sale of tobacco products to minors were observed. While policy implementers are ready to enforce legislation, the lack of a robust inspection system and weak enforcement, coupled with retailers' lack of awareness, pose significant barriers. Despite the government's reliance on tobacco products as a revenue source, strengthening inspections mechanism, enhancing legal authority, and raising public awareness are critical to overcoming these obstacles and ensuring effective policy execution.

Lastly, it is recommended that school authorities should be oriented and equipped to regulate tobacco sale as well as consumption near schools by coordinating with authorized officials. The local-level authorities should initiate the orientation of the retailers on legal provisions as well as enforce the rules for tobacco control along with a health promotion campaign for the students. At the same time, along with the current provisions of penalty in case of violation the TPCRA has to be revised as per current federal context.

## Supporting information

**S1 Data.**
(SAV)

**S1 Checklist.**
(DOCX)

## Author contributions

**Conceptualization:** Arjun Poudel, Hari Prasad Kaphle.

**Data curation:** Arjun Poudel, Sunita Poudel, Sagar Parajuli, Kumud Bhurtel.

**Formal analysis:** Arjun Poudel, Sunita Poudel, Sagar Parajuli, Kumud Bhurtel.

**Investigation:** Kumud Bhurtel.

**Methodology:** Arjun Poudel, Hari Prasad Kaphle, Shalik Ram Dhital.

**Software:** Arjun Poudel, Sunita Poudel, Sagar Parajuli.

**Supervision:** Hari Prasad Kaphle, Shalik Ram Dhital.

**Validation:** Hari Prasad Kaphle, Shalik Ram Dhital.

**Visualization:** Kumud Bhurtel.

**Writing – original draft:** Arjun Poudel, Sunita Poudel, Sagar Parajuli, Kumud Bhurtel, Shalik Ram Dhital.

**Writing – review & editing:** Arjun Poudel, Hari Prasad Kaphle, Shalik Ram Dhital.

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
