## [Decision Letter · Decision Letter 0]

PGPH-D-25-00229

Awareness and compliance with tobacco control policies among retailers nearby school in Arghakhanchi district of Nepal: A mixed method study

Dear Dr. Poudel,

Thank you for submitting your manuscript to PLOS Global Public Health. After careful consideration, we feel that it has merit but does not fully meet PLOS Global Public Health’s publication criteria as it currently stands. Therefore, we invite you to submit a revised version of the manuscript that addresses the points raised during the review process.

We look forward to receiving your revised manuscript.

Kind regards,

Chandrashekhar T. Sreeramareddy

Academic Editor

Journal Requirements:

Additional Editor Comments (if provided):

Reviewers' comments:

Reviewer's Responses to Questions

**Comments to the Author**

1. Does this manuscript meet PLOS Global Public Health’s publication criteria?

Reviewer #1: Yes

Reviewer #2: Yes

Reviewer #3: Yes

Reviewer #4: Yes

Reviewer #5: Yes

2. Has the statistical analysis been performed appropriately and rigorously?

Reviewer #1: Yes

Reviewer #2: Yes

Reviewer #3: I don't know

Reviewer #4: I don't know

Reviewer #5: Yes

3. Have the authors made all data underlying the findings in their manuscript fully available (please refer to the Data Availability Statement at the start of the manuscript PDF file)?

Reviewer #1: Yes

Reviewer #2: Yes

Reviewer #3: Yes

Reviewer #4: Yes

Reviewer #5: Yes

4. Is the manuscript presented in an intelligible fashion and written in standard English?

Reviewer #1: Yes

Reviewer #2: No

Reviewer #3: No

Reviewer #4: Yes

Reviewer #5: Yes

Reviewer #1: Please see attachment for numbered and highlighted review

PAPER REVIEW- COMMENTS

Title: Awareness and compliance with tobacco control policies among retailers of nearby schools in Arghakhanchi district of Nepal: A mixed methods study

Short title: Consider above

Order of authors: Arjun Poudel, Master's in public health- Remove apostrophe. Is your degree pluralized?

Abstract: The phrase ‘’retailers nearby school’’ is ambiguous and seems to run through your prose. Please reconsider

Consider ‘’barriers to the implementation of legislation at the district level’’ in place of barriers to implement the legislation…

25: close to a school OR close to schools

28: Could your estimated aOR be considered insignificant considering the CI; Clarify why you consider this worthy of noting please

Also were there any other significant inferences you could state to buttress your other findings?

39: found to be…

42 and school level students are growing of 42 cigarette smoking; Reconsider this phrase

Also the whole sentence is verbose and ambiguous. Kindly restructure

49: Reconsider scientifically

51, 52: Please justify with examples what exactly the tobacco companies have done with regards to interference with legislation?

55: ‘this study’ to replace ‘the current study’

56: Consider omitting and they

57: the district level

62, 63: Any reason your sites are in brackets?

64: You are missing a period

66: …and school administrators…

69, 70: Restructure sentence

79: remove according

80: Either examined by experts or experts were consulted; Rephrase sentence

97 Review grammar. Only 3 provisions those are directly?

112. Review …and most of them selling…

113 …were conviced…

116: Despite of that? Review please

120: See above

124: Apply above in a whole prose

137-140: Refer to comment 28; Also why was the odds and CI not stated for the other finding?

146: …properly implement…; Also review the whole sentence for grammatical errors

155,157: The examples you have cited as per countries do not substantiate that the problem is more severe in developing countries. India is still a developing country albeit a rapidly growing economy?

160: small amount? Reconsider please

161: one of the cities…

Generally a promising addition to the knowledge database, however requires meticulous fine-tuning.

Reviewer #2: The paper is clearly written and tackles a highly pertinent and timely issue. The authors have done an excellent job of addressing the key points and offering insightful perspectives on the topic. The writing is clear, and the thoroughness of the analysis makes it a valuable addition to the field. Well done!"

Reviewer #3: 1) Consider using destigmatized language, such as "people who smoke or use tobacco," instead of "tobacco users."

2) Strengthen the Introduction by including more data on the impact of tobacco use within the population and the implications of this work. Clearly articulate why this study is needed.

3) Review the manuscript for English grammar and clarity to improve readability.

In the Data Collection Tools and Procedure section:

1) Provide references to support the validity of the data collection tools and examples of questions or tools.

2) Include a more detailed description of the data analysis procedures—what steps were taken?

3) Specify how many researchers were included to ensure the validity of the conclusions for the qualitative data interpretation.

Reviewer #4: Good research question.

Please note the following:

1. Please use updated tobacco related mortality figure for Nepal- Kaur J, et al. Tob Control 2024;0:1–4. doi:10.1136/tc-2024-058599

2. It will be useful to mention prevalence of tobacco use among school going children in Nepal (GSHS 2015) (can be mentioned as a recommendation that a fresh survey is needed- GYTS/GSHS, as availability and accessibility of tobacco products is linked to use.

3. It will be useful to define retailers who sell tobacco products in general. Why only two types of retailers were taken- groceries/eateries? Is there other type of retailers too?

4. There is no mention about the law on ban of smokeless tobacco in public places.

5. Please mention about the source of funding if any

6. Limitations of the methodology used- please add

7. Please add some recommendations in the conclusion for different stakeholders including the school authorities, as they should report the retail outlets located so near to the school

Reviewer #5: The study addresses an important public health issue—tobacco control policy awareness and compliance among retailers near schools in Nepal. This topic is highly relevant to global public health, tobacco regulation, and adolescent health and contributes to the discussion on policy implementation in low- and middle-income countries. While studies on tobacco policy compliance exist in other countries, the localized focus on Nepal, particularly in school-adjacent retailers, provides a novel contribution.

Methodology: The sampling criteria for key informant interviews (KII) need further justification. Why were these specific officials chosen? Were any excluded, and if so, why? It would be more worthwhile if the the inclusion/exclusion criteria for retailers should be expanded beyond proximity to schools to account for factors such as store type or previous regulatory infractions in which are believed also available nearby school. However, if these factors are not included, further explanation will be useful as additional information for discussion.

Results and Discussion: The study mentions higher compliance in rural areas, but the underlying reasons should be explored further, not just acknowledging their less access to information. Are rural retailers more compliant due to greater community surveillance, lower competition, or stronger enforcement? A comparative discussion with similar studies from other regions would enhance the study’s contribution to the broader literature. How do compliance levels in Nepal compare to other low- and middle-income countries? It would benefit from comparing its justification to carry out the research, including the findings with similar studies in other regions (Southeast Asia, like Indonesia), not just with Kenya.

Policy Implications: While the study identifies enforcement and public awareness as facilitators, more actionable policy recommendations in addition discussing long-term impacts of non-compliance on public health outcomes are needed. Such as statements offering specific interventions to improve compliance. For example, could incentives or penalties improve compliance? Would community-based interventions help strengthen enforcement?

**Do you want your identity to be public for this peer review?** For information about this choice, including consent withdrawal, please see our Privacy Policy

Reviewer #1: **Yes: ** Dr Dr Nana Mireku-Gyimah

Reviewer #2: No

Reviewer #3: No

Reviewer #4: No

Reviewer #5: No

---

## [Decision Letter · Decision Letter 1]

Awareness and compliance with tobacco control policies among retailers near schools in Arghakhanchi, Nepal: A mixed- methods study

PGPH-D-25-00229R1

Dear Mr. Poudel,

We are pleased to inform you that your manuscript 'Awareness and compliance with tobacco control policies among retailers near schools in Arghakhanchi, Nepal: A mixed- methods study' has been provisionally accepted for publication in PLOS Global Public Health.

Best regards,

Chandrashekhar T. Sreeramareddy

Academic Editor

Reviewer Comments (if any, and for reference):

Reviewer's Responses to Questions

**Comments to the Author**

Reviewer #3: All comments have been addressed

Reviewer #5: All comments have been addressed

publication criteria?

Reviewer #3: Yes

Reviewer #5: Yes

3. Has the statistical analysis been performed appropriately and rigorously?

Reviewer #3: I don't know

Reviewer #5: Yes

4. Have the authors made all data underlying the findings in their manuscript fully available (please refer to the Data Availability Statement at the start of the manuscript PDF file)?

Reviewer #3: Yes

Reviewer #5: Yes

5. Is the manuscript presented in an intelligible fashion and written in standard English?

Reviewer #3: Yes

Reviewer #5: Yes

Reviewer #3: No additional comments.

Reviewer #5: In my view, the revised version of this paper shows significant improvement and has effectively incorporated the reviewers' feedback, thereby strengthening the overall argument regarding the importance of the research findings to advocate the urgent need for stricter enforcement of regulations due to the low awareness and compliance among retailers concerning the ban on smoking near schools in Nepal.

**Do you want your identity to be public for this peer review?** For information about this choice, including consent withdrawal, please see our Privacy Policy

Reviewer #3: No

Reviewer #5: No
